# The History and Development of Hyperbaric Oxygenation (HBO) in Thermal Burn Injury

**DOI:** 10.3390/medicina57010049

**Published:** 2021-01-08

**Authors:** Christian Smolle, Joerg Lindenmann, Lars Kamolz, Freyja-Maria Smolle-Juettner

**Affiliations:** 1Division of Plastic, Aesthetic and Reconstructive Surgery, Division of Thoracic and Hyperbaric Surgery, Medical University Graz, Auenbruggerplatz 29, A-8036 Graz, Austria; christian.smolle@medunigraz.at (C.S.); lars.kamolz@medunigraz.at (L.K.); 2Division of Thoracic and Hyperbaric Surgery, Medical University Graz, Auenbruggerplatz 29, A-8036 Graz, Austria

**Keywords:** hyperbaric oxygenation, history, review, burn injury

## Abstract

*Background and Objectives:* Hyperbaric oxygenation (HBO) denotes breathing of 100% oxygen under elevated ambient pressure. Since the initiation of HBO for burns in 1965, abundant experimental and clinical work has been done. Despite many undisputedly positive and only a few controversial results on the efficacy of adjunctive HBO for burn injury, the method has not yet been established in clinical routine. *Materials and Methods:* We did a retrospective analysis of the literature according to PRISMA—guidelines, from the very beginning of HBO for burns up to present, trying to elucidate the question why HBO is still sidelined in the treatment of burn injury. *Results:* Forty-seven publications (32 animal experiments, four trials in human volunteers and 11 clinical studies) fulfilled the inclusion criteria. Except four investigators who found little or no beneficial action, all were able to demonstrate positive effects of HBO, most of them describing less edema, improved healing, less infection or bacterial growth and most recently, reduction of post-burn pain. Secondary enlargement of burn was prevented, as microvascular perfusion could be preserved, and cells were kept viable. The application of HBO, however, concerning pressure, duration, frequency and number of treatment sessions, varied considerably. Authors of large clinical studies underscored the intricate measures required when administering HBO in severe burns. *Conclusions:* HBO unquestionably has a positive impact on the pathophysiological mechanisms, and hence on the healing and course of burns. The few negative results are most likely due to peculiarities in the administration of HBO and possibly also to interactions when delivering the treatment to severely ill patients. Well-designed studies are needed to definitively assess its clinical value as an adjunctive treatment focusing on relevant outcome criteria such as wound healing time, complications, length of hospital stay, mortality and scar quality, while also defining optimal HBO dosage and timing.

## 1. Introduction

### 1.1. History of Hyperbaric Oxygenation

In 1662, Henshaw, a British physician first utilized hyperbaric therapy, placing patients in a steel container that was pressurized with air. Though John Priestly discovered oxygen as soon as 1775, the marginally effective compressed air therapy was only cautiously replaced by breathing of 100% oxygen under increased ambient pressure, thus initiating “hyperbaric oxygenation”. The reason for the delay was the fear of side effects based on the work of Lavoisier and Seguin who had suspected toxic effects of highly concentrated oxygen in 1789. It took almost 100 years until in 1878 Paul Bert, who is considered the “father of the hyperbaric physiology”, documented the toxic effects of hyperbaric oxygen on the central nervous system that were manifested as seizures [1]. Yet, his findings took time to settle in the hyperbaric medical community. About half a century later in 1937, Behnke and Shaw first used hyperbaric oxygen successfully for the treatment of decompression sickness. In 1955, Churchill-Davidson [2] applied HBO to potentiate the effects of radiation therapy in cancer patients, while at the same time Boerema developed HBO as an adjunct to cardiac surgery, thus prolonging the time for circulatory arrest [3]. Since that time, HBO has been applied for a variety of medical conditions, as the pathophysiological and molecular mechanisms of hyperbaric oxygen treatment were increasingly understood.

### 1.2. Principle and Mechanisms of Hyperbaric Oxygenation

HBO denotes breathing of 100% oxygen under elevated ambient pressure between 2 and 3 atmospheres absolute (ATA) in a hyperbaric chamber. In direct correlation to the pressure level, oxygen physically dissolves in the plasma increasing arterial *p*O_2_. At a pressure of 2 ATA oxygen dissolves in the plasma resulting in an arterial *p*O_2_ of about 1400 mmHg, which can be further raised to 2000 mmHg at a pressure of 3 ATA. At 3 ATA, the sheer amount of dissolved oxygen obviates the need for erythrocytes for oxygenation [4]. Additionally, tissue oxygen tensions rise in accordance to arterial oxygen pressure and elevated levels may persist for several hours [5]. However, the mechanism of action of HBO is not mere hyper-oxygenation counteracting tissue hypoxia but is based on the fact that hyperbaric oxygen is a highly potent drug.

HBO redistributes blood flow causing vasoconstriction in regions with increased perfusion and vasodilation in hypoxic ones. On the molecular level HBO effectuates preservation of ATP, downregulation of complex molecular cascades involving ß-2 Integrin and pro-inflammatory cytokines, upregulation of anti-inflammatory cytokines and growth factors as well as mobilization of stem cells. Since microorganisms are unable to compensate for the high levels of oxygen, HBO exerts an unspecific antibacterial action. In addition, a reduction in leukocyte chemotaxis and an increase of phagocytosis enhance the efficiency of antibiotic treatment [6,7,8]. While problems in the middle ear and the nasal sinuses may be encountered during pressurization if there is obstruction due to swelling, side effects of the hyperbaric oxygen (paraesthesia, seizures) are very uncommon, if a pressure of 3 ATA is not exceeded. Even if they occur, they are quickly reversible if hyperbaric oxygen is switched to pressurized air [9].

### 1.3. HBO in Burn Injury

The use of HBO in burns was based on a serendipitous finding. In 1965, Japan, Wada and Ikeda [10] applied HBO treatment for severe CO intoxication to a group of coal miners who had also sustained second-degree burns during an explosion. In the HBO-treated miners the burns healed remarkably better than in other victims. Since then, HBO for burns has been dealt with in experimental and clinical trials and in numerous reviews [11,12,13,14,15,16].

When delving into the history of HBO for burn injury reviewing experimental and clinical work, one finds a considerable heterogeneity of study designs, and of injury characteristics such as type, extent, and depth as well as a variety of different species used in experimental settings. Additionally, the dose of HBO deriving from the factors magnitude of pressure, duration of the individual treatment session and total number of sessions varies considerably [17,18,19], as does the interval between the burn injury and the first HBO session. Since downregulation of mediator cascades is most effective if done as early as possible, this timespan has proved to be a crucial parameter in a variety of other indications [14,20,21,22].

We established a synopsis about the original animal and human experimental or clinical studies on HBO in burns published since 1965.

## 2. Materials and Methods

### Literature Search and Evaluation

We proceeded according to PRISMA guidelines [23]. For the terms “hyperbaric oxygen” and “burn” dating back to 1965, 314 articles were identified in Pubmed, 15 in Embase advanced and 5 in Cochrane databases. In addition, we found six relevant publications in proceedings of large international hyperbaric meetings. We included only publications the full-text of which was available. We excluded papers not providing sufficient information and redundant work. (For the PRISMA selection process, see Figure 1).

We evaluated species, number of individuals, type of study, % of total body surface area (TBSA), depth of burn, and pressure applied during HBO. As outcome parameters we documented metabolic effects, edema formation/fluid requirement, inflammation, micro-vessel patency/regrowth, infection, scarring, epithelization, pain, requirement for surgery, morbidity, mortality, duration of in-hospital stay and cost.

For descriptive statistics, each experiment was recorded as a statistical entity.

## 3. Results

### 3.1. General Considerations

Forty-seven publications (32 animal studies, four trials in human volunteers and 11 clinical studies or case series) fulfilled the inclusion criteria.

The total number of animals amounted to more than 3000, while there were 58 human volunteers and clinical studies in 2208 patients. Animal experiments were based on complex designs with up to 15 arms. On the contrary, only one out of four human volunteers [13] and two out of 11 clinical studies were prospectively randomized [24,25]. Three volunteer studies [26,27,28] used a crossover design, three clinical studies included matched pairs [29,30,31], four non-randomized controls [12,32,33,34] and two were case series [35,36].

The HBO regimens (pressure in ATA, duration of single session, frequency per day, total number of sessions) differed considerably. In clinical studies the interval between the injury and the first HBO was also inconsistent, and some authors gave no or incomplete information about the abovementioned factors. The same was true for the type of local treatment.

For the details of both experimental and clinical studies see Table 1 and Table 2, for the descriptive statistics see Table 3 and Table 4.

### 3.2. Animal Studies

#### 3.2.1. First Decade (1966–1977)

Marchal [37], Nelson [38] and Ketchum [39,40] did the first experimental trials in 1967 [39] followed by Benichoux in 1968 [41], Perrins [42] and Spinadel [43] in 1969, Gruber in 1970 [44], Ketchum again in 1970 [40], and Bleser in 1971 [45] and 1973 [46]. The numbers of animal were impressive amounting up to 520 per study. The investigators focused on both tolerance of HBO at a maximum of 3 ATA in general and influence of HBO on survival following large burns with a total body surface area (TBSA) of 75%. In 25–30% TBSA they studied the effect on healing. Härtwig in 1974 [47], Korn [48], Niccole [49], and Wells [50], all in 1977, investigated burns up to 40% TBSA and applied HBO at a maximum of 2.5 ATA focusing on healing and revascularization. Except Perrins and Wells all researchers did the experiments in rodents, predominantly rats.

Marchal [37] and his co-worker Benichoux [41] found prolonged survival when HBO was given in addition to fluid resuscitation with tris-hydroxymethyl aminomethane (THAM) buffer, in 75% TBSA burns, whereas HBO at 3 ATA without adjunctive treatment proved detrimental. Even unburnt rats tolerated HBO at 3 ATA poorly. Bleser [45,46], from the same group, repeated the experiments in 30% TBSA and documented less fluid loss and reduced fluid requirements as well as higher survival rates following HBO at 3 ATA combined with THAM and antibiotics, thereby also confirming the findings of Nelson [41] who had described a lesser drop of hematocrit following 75% TBSA burn and HBO at 2ATA. In smaller burns (TBSA 20–30%) Marchal was able to demonstrate less fluid loss, better granulation, faster healing, better quality of scars and less infection in the HBO group [37].

Ketchum, after inflicting burns up to 20% TBSA, reported similar results with reduction of both fluid requirements and edema, reduction of bacterial growth on burnt surfaces, lower incidence of sepsis, shortening of healing time by 30% and extensive capillary proliferation beneath the burn injury following HBO at 2 or 3 ATA [39,40].

Spinadel noted enhanced healing of 25% TBSA burns when applying HBO and antibiotics. He noted that guinea pigs seem to be susceptible to HBO at 3 ATA. There was no effect on mortality [43]. Gruber reported a marked increase of tissue-pO2 beneath burnt surfaces following 3 ATA HBO [44].

With HBO at 2.5 ATA in rats following <10% TBSA burns, Härtwig was another investigator to find quicker revascularization, less fluid loss, earlier shedding of scabs with wound healing occurring 6 days earlier than in controls [47]. Similarly, Korn [48] who investigated the effect of 2 ATA HBO in small burns of 5% TBSA in a large series of 211 guinea pigs noticed quicker epithelization, no full-thickness conversion, earlier return of vascular patency, and hardly any edema, loss of fluid or inflammation in HBO, while escars showed earlier shedding. Microangiography revealed rapid revascularization and viable cells beneath burnt areas in the HBO group, whereas mitotic activity in epithelia of controls was not evaluable due to widespread necrosis. Wells, the only one using dogs with 40% TBSA burns, found less reduction in plasma volume and less decline in postburn cardiac output in animals treated with 2 ATA HBO [50].

In contrast, Perrins [42], who was the only investigator using pigs in 12% TBSA experimental burn, documented hardly any local response to HBO treatment at 2 ATA. Of note, he applied HBO four times a day. Likewise, Niccole, when applying 2.5 ATA HBO in rats with 20% TBSA burns found neither difference in edema, time to epithelization nor to eschar separation. Yet, there was less bacterial colonization of burnt surfaces following HBO [49].

#### 3.2.2. Last Two Decades before Turn of the Century (1978–1998)

The investigators in this second phase used only rodents, again predominantly rats. Except Kaiser in 1985 [51] who did HBO at 3 ATA and Espinosa (2.8 ATA) [52], pressures never exceeded 2.5 ATA. While many studies corroborated the findings from earlier ones, new aspects were investigated, such as thrombosis, secondary enlargement of burn, mesenteric bacterial colonies, immunological effects and distinct histological changes within the burnt area.

Arzinger-Jonasch (1978) confirmed previous findings of a reduction in healing time for both full- and partial-thickness 15% TBSA burns and added the evidence that 2 ATA HBO prevented thrombosis beneath burnt areas and entailed a quicker take of full-thickness skin grafts [53]. Nylander (1984) showed a marked reduction of local edema by 2.5 ATA HBO after inflicting a burn on one mouse ear [54]. Kaiser, in 1985, studied the effect of HBO when treating < 10% TBSA burns infected by pseudomonas aeruginosa. He was able to demonstrate that infected wounds following primary HBO healed quicker than infected controls, whereas HBO applied after delay had no positive effect on healing time [51]. In a further study, in 1989, he focused on the secondary enlargement of burn injury. Whereas the phenomenon was reproducible in controls, HBO consistently prevented secondary enlargement of < 10% TBSA burns which in addition showed less edema and a quicker healing [55]. Stewart in 1989 reported consistently higher tissue ATP-levels beneath burnt surfaces when applying HBO [56]. In the same year, Saunders [57] found that 2 ATA HBO improved microvasculature and hence preserved perfusion of dermal und subdermal vessels resulting also in less permanent collagen denaturation subjacent to burnt surfaces. Tenenhaus, in 1994, investigated 30% TBSA burns and found fewer mesenteric bacterial colonies following 2.4 ATA HBO, as villus length in these animals remained normal [58]. Espinosa (1995) same as other investigator reported an HBO (2.8 ATA)—induced reduction of postburn edema following 10% TBSA burn [52]. Hussman, in 1996 was the first to investigate immunologic effects of 2.5 ATA HBO: He described a downregulation of cytotoxic (OX8) T-cells to normal values on days 5 and 15 after 10% TBSA burn [59]. Germonpre, in 1996, studied the influence of 2 ATA HBO on histological features within the burnt area. He documented less subepidermal leucocyte infiltration, better preservation of basal membrane and of skin appendages [60]. The only one reporting negative results in this period was Shoshani (1998) who was unable to find differences in intralesional perfusion between HBO-treated (2 ATA) animals and controls by laser-flowmetry. Besides, he described significantly slower epithelization with HBO [61].

#### 3.2.3. New Millennium (2002–2019)

Again, all studies were done in rodents, and all but one [62] in rats. The investigators set their focuses on intestinal bacteria [63], prevention of inflammation and necrosis as well as on regeneration [62,64,65,66] and on alleviation of pain [67,68] induced by burn injury. The maximum pressure applied was 2.5 ATA.

Following 2.5 ATA HBO in 30% TBSA burn Akin [34] found lower bacterial colony counts in the distal ileum. Bacterial overgrowth and translocation to lymph nodes, blood liver and spleen were prevented. Bilic (2005) who inflicted 20% TBSA burn injury, confirmed former investigators’ results describing reduction of edema, and enhancement of neo-angiogenesis and epithelization. The latter was linked to a higher number of preserved regeneratory follicles [64]. Similarly, Türkaslan in 2010 reported reduced edema as well as preservation of vital cells and cells in the proliferative phase in 5% TBSA burn. Furthermore, HBO prevented progression from zone of stasis to necrosis [65]. Selcuk, in 2013, investigated the effect of HBO on 12% TBSA burn in rats, half of which had undergone pretreatment with nicotine. HBO reduced the degree of necrosis in these animals, while the least degree of necrosis, best epithelization, and lowest inflammatory cell response was present in rats who had HBO treatment and no nicotine [66]. Finally, Hatibie in 2019 was able to show fewer inflammatory cells and better epithelization as well as a trend for enhanced angiogenesis following HBO in 1% TBSA burns [62]. Wu as well inflicted small burns (1% TBSA) when studying wound-healing and postburn pain. Early HBO inhibited the Gal-3 dependent TLR-4 pathway, thus reducing proinflammatory cytokines and proteins both in the affected extremity and in the hind horn. Microglia and macrophage activation following burn injury were suppressed. This resulted in a decrease of the mechanical withdrawal threshold and in a promotion of wound healing (2018) [67]. In a recent study, in 2019, he was able to show, that HBO reduced burn-induced mechanical allodynia, in correlation to the duration of treatment. The effect was accompanied by an upregulation of melatonin and opioid receptors, and by downregulation of brain derived neurotropic factor, substance P and calcitonin gene-related peptide [68].

For detailed baseline data, specifications of HBO and results see Table 1, Table 2, Table 3 and Table 4.

### 3.3. Human Volunteers

The first study in human volunteers was done by Hammarlund in 1991 [26], who used a crossover design inflicting a 5 mm in diameter UV-blister suction wound on the forearm followed by HBO at 2.8 ATA. There was a significant reduction of exsudation and hyperemia as well as of wound size, after HBO but no effect on epithelization. Six years later, Niezgoda used an almost identical setup, yet in a randomized design applying 2.4 ATA HBO. His findings did not differ from those of Hammarlund [13]. In 2015, Rasmussen published another crossover study focused on postburn pain, inflicting a superficial burn with a thermode whereafter HBO at 2.4 ATA was applied. The treatment attenuated the central sensitation by thermal injury. A peripheral inflammatory effect was ruled out [27]. Wahl, from the same group, used an identical setup in 2019. After a single HBO treatment, an immediate mitigating effect on hyperalgesia followed by a long-lasting reduction of pain sensitivity surrounding the injured area was documented [28].

### 3.4. Clinical Studies

All clinical studies involved full and partial thickness burns. Their extent ranged between 35 and 80%. HBO was administered in addition to routine burn treatment. In contrast to the experimental work, where HBO was almost exclusively administered immediately following burn, this interval varied in the clinical settings: In two studies the patients were treated within 24 h [24,31], in one within 15 [25], in two within 12 [12,30], and in one within four hours after the incident [32]. Brannen stated a mean of 11,5 h with one third of patients treated within 8 h [31]. Five authors did not convey information on this parameter.

Ikeda, in 1967, was the first to report positive effects of 3ATA HBO in 43 extensively burnt (TBSA up to 90%, median 65%) patients after explosion in a mine. They had HBO for additional CO-intoxication and did better than anticipated and better than those who had no HBO treatment. Ikeda underscored the absence of superinfection or sepsis in the HBO collective [36]. In 1979 Lamy published a series of 27 patients with 50% TBSA burns who had 3 ATA HBO with intention to treat. Though mortality was not altered, Lamy noticed better healing and fewer infections [35]. Encouraged by these findings Hart applied 2 ATA HBO both with intention to treat and in a randomized study, published in 1974 [24]. He treated 138 patients with an average TBSA of 35% comparing them to 53 historical controls. Time to healing, morbidity, and mortality decreased beyond the values to be expected in each at risk group. The randomized patients (four with 15%, 25%, 35% or 45% TBSA each) showed reduced mean healing time in relation to %TBSA when compared to the controls. Grossmann administered 2 ATA HBO to patients with an average of 40% TBSA, incorporating the method into routine. He compared 419 patients with routine treatment to 419 who had additional HBO. Fluid requirements, healing time in second-degree burns, eschar separation time in 3rd degree, donor graft harvesting time, length of hospital stay, complications and mortality were all reduced compared to the non-HBO group. In addition, HBO seemed to prevent paralytic ileus in severe burns. In spite the additional treatment Grossman documented a reduction in overall cost [32].

In 1982, Waisbren designed a matched pairs study with 36 patients each, comparing routine treatment to 2.5 ATA HBO in 50% TBSA burn. In contrast to the other investigators, he observed worse renal function, lower rate of non-segmented, polymorphonuclear leucocytes and a higher rate of bacteriemia in the HBO-treated group. Yet, he confirmed former findings of improved healing, resulting in 75% lower requirement of skin grafts in the HBO group [29]. In 1987, Niu published a study comparing 226 patients with 2.5 ATA HBO treatment to 609 historical controls. TBSA was 35%. He added further proof to the evidence of reduced fluid loss and earlier re-epithelization. Yet, the overall mortality was not different from the one of the controls, though it was lower than in controls if only the high-risk group was considered [33].

Cianci and his group in 1989, who similar as Grossman had established HBO as a treatment routine for burn injury, described a small collective of 20 patients with 30% TBSA comparing HBO at 2ATA to historical controls. He found both duration of hospitalization and number of necessary surgical interventions reduced by HBO treatment [12]. A further study in 1990, this time using a matched pairs design in 21 patients, confirmed the former findings. In addition, Cianci stated a reduction of cost in burn care in the HBO-group thereby confirming Grossman´s results [30].

Brannen, in 1997, did a matched-pairs study in 125 patients, 63 of whom had 2 ATA HBO. While renal insufficiency seemed more frequent in the HBO-group, he was unable to document any difference in number of surgeries, length of hospital stay or mortality. On the other hand, Brannen mentioned less fluid loss following HBO. The information about TBSA and details of HBO—application was scant [31]. Chong, in 2013, randomized 8 patients with 13% TBSA burns to 2.4 ATA HBO and 9 to the control group. He found no effect on inflammatory markers IL Beta, 4, 6 and 10, yet a significantly lower rate of positive bacterial cultures of staphylococcus aureus and pseudomonas aeruginosa following HBO [25]. Finally, Chen compared 17 historical controls to 18 patients with 25% TBSA burn who had HBO at 2.5 ATA. Though the post-burn pain scores were better in the HBO group, he noticed no effect on infection or quality of scarring [34].

## 4. Discussion

We established a comprehensive synopsis of the experimental and clinical work on HBO in burns from its beginning in 1965 to the present, step by step addressing the crucial pathogenetic factors in burns which result from a combination of direct tissue damage by the thermal trauma and initiation of mediator cascades. The latter cause edema, coagulopathy, microvascular stasis and secondary enlargement of the local damage and systemic inflammatory reaction. In large burns, intestinal dysfunction results from both systemic inflammation and imbalance of intestinal microorganisms. Superinfection of burnt surfaces is a further threat [69].

Both experimental and clinical work throughout five decades has yielded unquestionable evidence of beneficial HBO effects the abovementioned factors. Early experiments relied on simple methods of description, whereas later on the evaluation became more sophisticated as insight into underlying molecular mechanisms of HBO developed [8].

Promotion of healing was demonstrable in both partial- and full-thickness burns. It involved quicker epithelization and thus more rapid healing of burnt surfaces based on prevention of secondary enlargement of the injury, preservation and better restoration of the microvasculature and the skin appendages as well as higher levels of intracellular ATP in the burnt area. Earlier shedding of escars was observed in full-thickness burns. Only three out of 32 experimental [42,49,61] and one out of 11 clinical studies [31] did not confirm positive effects of HBO on healing.

An undisputed phenomenon was the marked reduction of post-burn edema and of fluid requirements, respectively [13,26,32,37,38,45,47,48,50,52,53,54,55,64,65]. Likewise, inhibition of bacterial growth on both burn wounds and in the intestinum, resulting in fewer cases of sepsis [36] in an absence of bacterial translocation [63] and even in a lower incidence of ileus [32], was confirmed by all investigators focusing on that very issue [25,32,35,36,39,46,49,58,63].

Recent investigations focusing on postburn pain documented marked reduction of sensitation that could be induced by even one single HBO treatment session [27,28,34,67,68].

The findings about the impact of HBO on the prognosis of experimental or clinical burns, respectively, were less consistent: Marchal [37], Benichoux [41] and Bleser [45] reported prolonged survival or less mortality, respectively, in 30 to 75% TBSA burns only when HBO was combined with fluid substitution and buffering or antibiotics, respectively. Spinadel, on the contrary, found no effect on mortality in 25% TBSA experimental burns [43]. From the clinical point of view, Grossmann [32] and Hart [24] reported generally reduced mortality, while Grossmann also noticed fewer complications and fewer surgical interventions in the HBO patients, resulting in shortened hospital stay and lower cost [32]. Cianci confirmed these findings [30]. Niu observed better survival in the high-risk group [33], yet documented an unaltered course in less extensive burns [33]. In contrast, Lamy found prognosis unchanged in extensive burns [35], and Brannen described survival generally unchanged [31].

The reasons for these disparities are most likely based on a variety of factors.

First, the effects of HBO, which basically constitutes a kind of pharmacological treatment, are dose-dependent. The dose of HBO, however, derives from the combination of pressure, duration of exposure, frequency of treatment and total number of treatments—in other words, there is no “HBO” as such. In consequence, all the factors contributing to the HBO dose differed largely between studies and hardly any identical treatment regimens could be identified. What is more, the information about the dose-defining parameters of HBO was incomplete in many clinical studies. The only explicit finding was the gradual use of lower pressure levels. While 3 ATA were applied in the early years of research, there was a general tendency to apply lower pressures as time went by. Evidence of overdose—effects due to high pressure were shown in rats [37,46] and guineapigs [43] at 3 ATA, whereas Tenenhaus noticed negative impact of duration and frequency in mice when applying 2.4 ATA throughout 120 min three times a day [58]. The lack of positive effects of 2 ATA for 60 min used by Perrins may also have been influenced by the fact, that he applied the treatment four times a day [42]. A study on the impact of HBO on bone repair [70] proved that one application per day yielded better results than two and supports this assumption.

In any acute injury HBO has the potential to downregulate mediator cascades if applied within the appropriate time window. In stroke [22] and acute carbon monoxide intoxication [20], this interval seems to last until 6 to 8 h after the acute event. Later on, the inflammatory cascade is less likely to be downregulated by HBO and after 24 h this line of action may be terminated. These limitations probably also apply to HBO in burn injury [18,19,51,54]. The available data from clinical studies are too sparse as to provide an answer to this question.

In summary, though there are recommendations to apply a maximum of 2.5 ATA, there is still no generally established HBO treatment regimen for adjunctive HBO in burns [19].

The delivery of HBO to intensive care patients with precarious fluid balance and requirement for vasopressor treatment and artificial ventilation is a critical issue. The necessary repeated transfer of patients into the chamber for hyperbaric treatment involves further stress and the mere logistics of HBO—delivery may deter therapists from its use. Though HBO can be administered safely and with beneficial effect to severely ill patients, the treatment requires a high degree of experience in hyperbaric medicine and can be demanding for the therapist. Infusion rates, vasopressor doses and ventilation parameters need to be repeatedly adjusted during each HBO session and a meticulous invasive monitoring of cardiorespiratory parameters is mandatory. Thus, in unexperienced hands, HBO treatment may fail to succeed or may even have adverse effects in burn victims [16,32,71,72], a fact that may explain some negative results. The 2018 European Committee for Hyperbaric Medicine (ECHM) consensus paper stresses the point, that HBO treatment in burn should only be administered in burn centers with direct access to highly specialized HBO units [19].

What is more, both experimental and especially clinical studies on burn treatment used different outcome issues or different definitions, respectively, to assess the same outcome issues, while results were evaluated at different time points after the injury [71,73]. In addition, like all reviews of clinical studies in burns, we had to face the heterogeneity of basic parameters including epidemiology, comorbidity, as well as type, degree, TBSA and localization of the burn. Young and colleagues, in a recent systematic review of burn treatment, suggested the development of a core outcome set (COS) enabling the comparison of results of studies on burn treatment [73]. This basic problem when designing trials in burn patients applies also for HBO: Only two clinical studies were randomized including a total of 16 [24] and 15 patients [25], respectively. Other investigators relied on matched pairs [29,30,31] which allowed for a maximum number of 62 per group. When evaluating the divergent results also the low numbers of patients must be considered.

The main limitation of the study is the lack of comparability of both the various experimental and clinical setups and treatment schedules. In addition, relevant data were not consistently available in all publications.

## 5. Conclusions

Much experimental and clinical research has been done on the topic of HBO in burn injury, and the authors provide well founded evidence of beneficial interaction with the pathomechanisms of burns and healing processes, albeit focusing on single issues in the various investigations [71]. A comprehensive experimental view on the timeline of patho-molecular events and their interaction with HBO treatment would be desirable. To definitively assess its hitherto disputed clinical value as an adjunctive treatment, however, there is a dire need for well-designed clinical studies. They should involve relevant outcome criteria such as wound-healing time, complications, length of hospital stay, mortality and scar quality, while also defining optimal HBO dose and timing. They will have to be conducted in highly specialized burn treatment units equipped with hyperbaric treatment facilities and expert staff familiar with hyperbaric intensive care. As there is hardly any other single therapeutic measure by which—at least in experimental use—so many aspects of burn can be dealt with, intensified research on this issue is worthwhile.

## Figures and Tables

**Figure 1 medicina-57-00049-f001:**
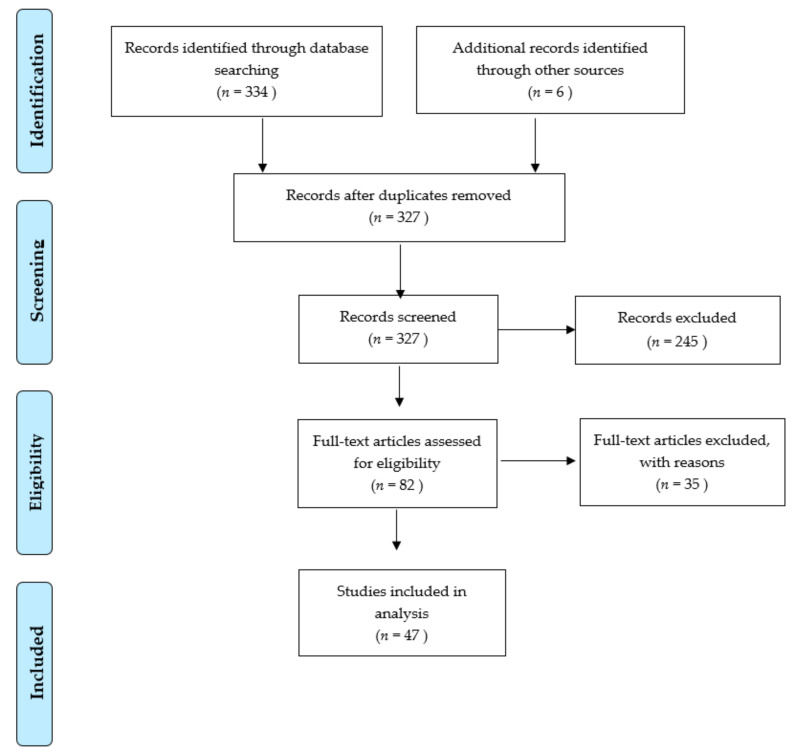
PRISMA selection process.

**Table 1 medicina-57-00049-t001:** Depth of burns: PT: partial thickness, FT: full thickness, S: superficial; HBO: hyperbaric oxygenation; NBO: normobaric oxygenation; THAM: Tris (hydroxymethyl) aminomethane buffer.

Author/Year	Species	Nr. Individuals	Study Design	% TBSA	Depth of Burn	Local Treatment
Marchal/1966	Rats	187	no burn/HBO: 10			
			burn/untreated: 15; burn/HBO: 15	20	FT	
			7 arm (at least 21 each;) burn/no treatment burn/HBO only burn/saline burn/saline, HBO burn/THAM burn/THAM, HBO (each day) burn/THAM, HBO (every other day)	75	PT	
Nelson/1966	Dogs	24	2 arm; burn/untreated: 12 burn/HBO: 12	75	PT	
Ketchum/1967	Rabbits	26	2 arm; burn/untreated: 13; burn/HBO: 13	5	FT, PT	
Ketchum/1970	Rats	30	2 arm; burn/untreated: 15 burn/HBO: 15	20	FT	
Benichoux/1968	Rats	160	8 arm; no burn/HBO: 10 burn/no treatment: 25 burn/HBO only: 25 burn/saline: 25 burn/saline, HBO: 25 burn/THAM: 25 burn/THAM, HBO every day: 25 burn/THAM, HBO every second day: 25	75	PT	
		200	8 arm; no burn/HBO; burn/THAM; burn/HBO every second day; burn/colmycine, penicilline; burn/THAM, HBO; burn/penicilline, colimycin, HBO; burn/THAM, penicilline, colimycine burn/THAM, penicillin, colimycine, HBO	30	PT	
Perrins/1969	Pigs	8	2 arm; burn/untreated: 4 burn/HBO: 4	12	FT	
		4	2 arm; burn/untreated: 2 burn/HBO: 2	8	PT	
Spinadel/1969	Guinea pigs	99	3 arm; burn/untreated: 33 burn/antibiotics 33 burn/HBO & antibiotics: 33	25	PT	Gentamycin-powder
	Hamsters	75	3 arm; burn/untreated: 25 burn/antibiotics: 25 burn/HBO & antibiotics: 25	25	PT	Gentamycin-powder
Gruber/1970	Rats	24	3 arm; pedicled flap, replanted/HBO: 8 composite skin graft, replanted/HBO: 8; burn/HBO: 8	10	FT	
Bleser/1971	Rats	520	3 arm; no burn/untreated: 20 burn/untreated: 250 burn/HBO, THAM, penicillin, colimycine: 250	32	PT	
Bleser/1973	Rats	100	2 arm; burn/untreated: 50 burn/HBO: 50	5	FT	
Härtwig/1974	Rats	100	2 arm; burn/untreated: 50 burn/HBO: 50	2	FT	
Korn/1977	Guinea pigs	117	Series I: 3 arm; burn/HBO: 52 burn/Hyperbaric normoxia: 27 burn/untreated: 38	5	PT	open/pro-tected
		54	Series II: 2 arm; burn/HBO: 30 burn/untreated: 24	5	PT	open/pro-tected
		40	Series III: 4 arm; no burn/control: 8 no burn/HBO: 8 burn/untreated: 12 burn/HBO: 12	5	PT	open/pro-tected
Niccole/1977	Rats	80	4 arm; burn/untreated: 20 burn/HBO: 20 burn/sliver-sulfadiazine: 20 burn/HBO/silver-sulfadiazine: 20	20	40 PT 40 FT	sulfadiazine (removed before HBO); no dressings
Wells/1977	Dogs	24	3 arm; burn/no fluid: 8 burn/no fluid, NBO: 8 burn/fluid, HBO: 8	40	FT	
Arzinger-Jonasch/1978	Guinea pigs	120	5 arm; burn/HBO: 10	15	PT	
			burn/HBO: 10		FT	
			burn/HBO: 20; necrectomy at various time points		FT	necrectomy
			burn/HBO: 20; necrectomy, full-thickness grafts at various time points		FT	necrectomy/skin graft
			burn/necrectomy, full-thickness grafts at various time time points: 60;		FT	necrectomy/skin graft
Nylander/1984	Mice	54	2 arm; burn/untreated: 27 burn/HBO: 27	6	PT	
Kaiser/1985	Guinea pig	102	5 arm: burn not infected/untreated: 21 burn not infected/HBO: 21 burn infected (pseudomonas)/untreated: 30 burn infected (pseudomonas)/primary HBO: 15 burn infected (pseudomonas)/secondary HBO: 15	5	FT	
Kaiser/1988	Guinea pigs	75	2 arm; burn/untreated: 43 burn/HBO: 32	5	PT	
Stewart/1989	Rats	90	15 arm; no burn/untreated: 6			
			burn/untreated, biopsy at 12 h: 6	5	PT	silver sulfadiazine
			burn/1 HBO, biopsy at 12 h: 6		PT	
			burn/2 HBO, biopsy at 12 h: 6		PT	
			burn/1HBO biopsy at 36 h: 6		PT	
			burn/untreated, biopsy; 5 groups of 6 animals each at 36, 48, 72, 96 or 120 h		PT	
			burn/2 HBO, biopsy; 5 groups of 6 animals each at 36, 48, 72, 96 or 120 h		PT	
Saunders/1989	Guinea pigs	30	2 arm; burn/untreated: 15 burn/HBO: 15 (3 different times of evaluation: 6, 24, 48 h)		PT	
Tenenhaus/1994	Mice	125	5 arm; no burn/fluid, food: 22 burn/fluid, food: 32 burn/fluid, food, compressed air: 15 burn/, fluid, food, NBO: 24 burn/fluid, food, HBO: 32	32	FT	
		139	4 arm; no burn/fluid, no food: 22 burn/fluid, no food: 51 burn/fluid, no food, HBO 2 × 120 min: 57 burn/fluid, no food, HBO 3 × 120 min: 9	32	FT	
Espinosa/1995	Guinea pigs	20	3 arm; burn/untreated: 6 burn/HBO: 7 burn/HBO, antibiotic: 7	10	PT	
Hussmann/1996	Rats	74	11 arm; no burn/untreated: 4	10	FT	
			no burn/anaesthesia, untreated: 7			
			burn/untreated: 7		FT	
			excision of 10% TBSA/suture: 7			excision
			no burn/HBO (acute): 7			
			no burn/HBO (chronical): 7			
			burn/excision after 4 h: 7		FT	excision
			burn/HBO (once): 7		FT	
			burn/HBO (repeated): 7		FT	
			burn/excision after 4 h and HBO (once): 7		FT	excision
			burn/excision after 4 h and HBO (repeated): 7		FT	excision
Germonpré/1996	Rats	46	3 arm; burn/untreated: 10 burn/HBO: 17 burn/Piracetam: 19	5	PT	mafenide gauze; Op-Site
Shoshani/1998	Guinea pigs	54	3 arm; burn/silversulfadiazine: 18 burn/NBO, silversulfadiazine: 18 burn/HBO, silversulfadiazine: 18	5	PT	silver sulfadiazine
Akin/2002	Rats	54	7 arm; no burn/liquids: 6 no burn/liquids, HBO (short): 8 no burn/liquids, HBO (long): 8 burn/liquids: 16 burn/liquids, HBO (short): 8 burn/liquids, HBO long): 8	30	PT	
Bilic/2005	Rats	70	2 arm randomized; burn/Hyperbaric normoxia: 35 burn/HBO: 35	20	PT	silver sulfadiazine
Türkaslan/2010	Rats	20	4 arm; burn/untreated (evaluation at 24 h): 5 burn/HBO (evaluation at 24 h): 5 burn/untreated (evaluation on day 5): 5 burn/HBO (evaluation on day 5): 5	5	PT	
Selcuk/2013	Rats	32	4 arm; burn/Nicotine, HBO: 8, burn/Nicotine: 8; burn/no nicotine/HBO: 8; burn/no Nicotine: 8	12	PT, FT	
Wu/2018	Rats	36	6 arm; *sham*-burn/sham HBO: 6 sham burn/HBO: 6 burn/1 week sham HBO: 6 burn/2 week sham HBO:6 burn/1 week HBO: 6 burn/2 weeks HBO: 6	1	FT	silver sulfadiazine
Wu/2019	Rats	30	5 arm; Sham-burn/Sham HBO: 6; sham burn/HBO: 6; burn/1 week sham HBO:6; burn/1 week HBO: 6; burn/2 weeks HBO: 6	1	FT	silver sulfadiazine
Hatibie/2019	Rabbits	36	2 arm; burn/untreated; 18 burn/HBO: 18	1	PT	vaseline
Ikeda/1967	Patients	43	case series	>50	PT, FT	silver nitrate 0,5%
Lamy/1970	Patients	27	case series, historical comparator	20 to >80	PT, FT	
Hart/1974	Patients	191	2 arm double blind randomized; (included in observational 138 burn/HBO: 138 and burn/sham HBO: 53)	10 to 50	PT, FT	silver sulfadiazine
			Group I (HBO: 2; sham HBO: 2)	>10 <20		
			Group II (HBO: 2; sham HBO: 2)	>20 <30		
			Group III (HBO: 2; sham HBO: 2)	>30 <40		
			Group IV (HBO: 2; sham HBO: 2)	<40 <50		
Grossmann/1978	Patients	821	2 arm; nonrandomized controls; burn/routine treatment: 419; burn/routine treatment & HBO: 421	>20 <80	PT, FT	silver sulfadiazine
Waisbren/1982	Patients	72	2 arm: matched pairs; burn/routine treatment: 36; burn/routine treatment & HBO: 36	about 50	PT, FT	
Niu/1987	Patients	835	2 arm; nonrandomized comparator; burn/routine treatment: 609; burn/routine treatment & HBO: 226	any; subgroup severe burns	PT, FT	
Cianci/1989	Patients	20	2 arm: nonrandomized controls burn/routine: 12 (had no access to HBO); burn/routine treatment & HBO: 8	18–39	PT, FT	
Cianci/1990	Patients	21	matched pairs burn/routine treatment: 11; burn/routine treatment & HBO: 10,	19–50 (mean: 30)	PT, FT	
Hammar-lund/1991	Volunteers	8	2 arm cross-over at 10-day interval burn/untreated: 8 burn/HBO: 8	<1	PT	polyurethane film or hydrocolloid
Brannen/1997	Patients	125	2 arm matched pairs; burn/routine treatment: 62 burn/routine treatment & HBO: 63	20–50 (mean)	PT, FT	
Niezgoda/1997	Volunteers	12	2 arm randomized; burn/NBO: 6 burn/HBO: 6	<1	PT	hydrocolloid
Chong/2013	Patients	17	2 arm randomized; burn/routine treatment: 9 burn/routine treatment & HBO: 8; non-intubated	<35 (mean: 13)	PT, FT	bio-occlusive dressing
Rasmussen/2015	Volunteers	17	2 arm crossover: burn/HBO–NBO: 17 burn/NBO–HBO: 17	1	S	
Chen/2018	Patients	35	2 arm retrospective case control; burn/routine treatment: 17 burn/routine treatment & HBO: 18	<60	PT, FT	
Wahl/2019	Volunteers	21	2 arm crossover; burn/HBO–NBO: 12 burn/NBO–HBO: 9	1	S	

**Table 2 medicina-57-00049-t002:** HBO treament features; summarized treatment results

Author/Year	Interval Burn—HBO (hours)	Pressure (ATA)	Duration min	HBO Frequency/Day	Total Number HBO Sessions	Results in Detail
Marchal/1966		3	60	once a day	21	1 rat died, one convulsed; no weight gain during treatment
		3	60	once a day	28	After day 12 better granulation, faster healing, less infection in HBO
		3	60	once a day or every second day	5 to 10	mortality with daily HBO alone higher than in untreated controls. best survival in THAM with HBO every second day
Nelson/1966	0.1	2	60	once	1	hematocrit drops less after HBO
Ketchum/1967		2	60	four times a day with 1 h in between for 23 days	92	healing time reduced by 30%; reduction in positive cultures by 50%, purulent infection reduced
Ketchum/1970		3	60	four times a day with 1 h in between for 28 days	112	angiography: after day 28 extensive capillary proliferation underneath burn in HBO group; Histology: abundant capillary plexus
Benichoux/1968		3	60	once a day	up to 10	positive effect on mortality in THAM and HBO
		3	60	every second day	up to 15	HBO alone has no effect on burn, may even have adverse effect on survival; in THAM improved survival; longest survival in HBO&THAM& antibiotics
Perrins/1969	2	2	60	four times a day	12	burns in HBO group healed slower than in controls
	2	2	60	four times a day	12	no effect of HBO on healing process, no effect on depth of slough.
Spinadel/1969		2	75			HBO & antibiotics best results concerning healing; HBO alone and antibiotics alone equal but less good; untreated controls do markedly worse.
		3	120			HBO & antibiotics best results concerning healing; HBO alone and antibiotics alone equal but less good; untreated controls do markedly worse.
Gruber/1970	24	3	45	once every week	3	return of pathologically low oxygen tensions to normal achieved by HBO in flaps, grafts or burns; oxygen levels returning to pretreatment values soon after discontinuing HBO
Bleser/1971	0.1	3	60	every second day	4	rapid restoration of total body water; hematocrit, blood volume, plasma volumein HBO; accerlerated recovery in HBO
Bleser/1973	0.1	3	60	once a day	28	first no effect, but soon better granulation, more rapid healing and less infection with HBO.
Härtwig/1974	0.3	2.5	60	three times a day	84	hardly any edema or inflammation in HBO, hardly any loss of fluid; earlier shedding of eschar; microangiography: rapid revascularization in HBO
Korn/1977	0.5	2	90	twice a day	6, 8 or 10	quicker epithelization, no full-thickness conversion in HBO
	0.5	2	90	twice a day	2, 4, 6, or 8	earlier return of vascular patency in HBO
		2	90	twice a day	8	mitotic activity in epithelia of burnt controls not evaluable due to widespread necrosis
Niccole/1977	12	2.5	90	twice a day	(75?)	no difference concerning edema for either FT or PT; no differences in treatement groups for time to epithelization in PT or to eschar separation in FT; less bacterial colonization in FT after HBO& sulfadiazine and in sulfadiazine alone
Wells/1977	0.5	2	60	once	1	less reduction in plasma volume in HBO
	0.5	3	60	twice	2	less reduction in plasma volume in HBO; less decline in postburn cardiac output in HBO
Arzinger-Jonasch/1978		2	60 or 120	once a day	10	Time until healing of partial or full-thickness burn shortened by 5 days in HBO. Quick reduction of edema, hardly any thromboses, collateral perfusion in HBO. Take of full-thickness skin graft shortened by 2 days in HBO. Positive effect unrelated to time of exposition.
Nylander/1984	<0.1	2.5	45	once a day	1	less local and general edema formation at 2, 6 and 24 h after burn (fluid content of ear post HBO similar to untreated one).
Kaiser/1985	1 in primary HBO; 192 in secondary HBO	3	60	3 times a day	up to 81 (until closure of wounds)	noninfected wounds in controls healed quicker than noninfected HBO treated wound; infected wounds treated with primary HBO healed quicker than infected controls; infected wounds treated with secondary HBO healed somewhat slower.
Kaiser/1988	1	3	60	3 times a day		Extent of burn increased in controls, not in HBO-group; Rapid reduction of wound surface and less edema only in HBO-group
Stewart/1989						
						consistently higher tissue ATP in HBO; 2 HBO/day better than one
	0.5	2.5	60	once a day	1	
	0.5	2.5	60	twice a day	2	
	0.5	2.5	60	once a day	1	
	0.5	2.5	60	twice a day	2 to 10	36 h post injury, with 2 HBO/day more than tenfold increase in tissue ATP compared to 36 h. controls
Saunders/1989	2	2	60	once a day	up to 4	HBO improved microvascularity in all groups; perfusion of dermal und subdermal vessels beneath burn preserved; less permanent collagen denaturation
Tenenhaus/1994	0.5	2.4	90 or 120	twice a day	2	mesenterial bacterial cultures are postburn sign. HBO: fewer mesenteric bacterial colonies; fewest colonies in fed, HBO treated mice; Villus length in HBO treated, fed mice as long as in nonburnt controls.
			120	twice a day or three times a day	3	fasting produced more bacterial colonies. Three 120 min HBO per 24 h had detrimental effects (seizures);
Espinosa/1995	1	2.8	60	twice a day	8	significant reduction of edema in HBO with or without antibiotic
Hussmann/1996						
		2.5	90	once	1	
		2.5	90	twice a day	up to 14	
	4	2.5	90	once	1	increase of cytotoxic cells unchanged
	4	2.5	90	twice a day	up to 14	increase of cytotoxic cells unchanged
	4	2.5	90	once	1	only regimen to prevent increase of cytotoxic (OX8) T-cells on day 1
	4	2.5	90	twice a day	up to 14	only regimen to downregulate cytotoxic (OX8) T-cells to normal values on days 5 and 15
Germonpré/ 1996	4	2	60	every 8 h first day, thereafter twice a day	6	Histology day 3: less subepidermal leucocyte infiltration, better preservation of basal membrane * and of skin appendages * after HBO; piracetam has effect only on basal membrane
Shoshani/1998	up to 24	2	90	once within first 24 h, twice a day thereafter	29	epithelization significantly slower with HBO
Akin/2002		2.5	90	twice a day	4	day 3: HBO prevents intestinal bacterial overgrowth and translocation to lymph nodes, liver and spleen
					14	day 8: HBO prevents bacterial overgrowth and translocation to lymph nodes, liver and spleen
Bilic/2005	2	2.5	60	once a day for up to 21 days	21	Skin samples day 1, 2, 3, 5, 7, 15, 21: less edema, increased neoangiogenesis, higher number of regenatory follicles, earlier epithelization; no significant difference in necrosis staging or margination of leucocytes
Türkaslan/2010	0.5	2.5	90	twice a day	2 or 10	no differences in the 24 h-groups; 5 day group HBO: Vital zones preserved; more cells in proliferative phase, more vital cells; prevents progression from zone of stasis to necrosis, less edema
					10	augmented neovascularization, decreased edema in HBO; no secondary enlargement of burn area
Selcuk/2013	1	2.5	90	once a day	7	After 21 days no difference concerning microbiology; yet best epithlization, lowermost inflammatory cell response, fewest fibrosis in non-nicotine/HBO
Wu/2018	24	2.5	60	once a day	5 or 10	early HBO inhibits Gal-3 dependent TLR-4 pathway; decreases proinflammatory cytokines and proteins in hind horn and paw; suppresses microglia/macrophage activation following burn injury; decreases mechanical withdrawal threshold; promotes wound healing;
Wu/2019	24	2.5	60	once a day	5 or 10	more HBO sessions reduce burn—induced mechanical allodynia (upregulation: melatonin, opioid-receptors, downregulation: brain derived neurotropic factor, substance P, calcitonin gene related peptide)
Hatibie/2019		2.4	90	once a day	6	day 14: fewer inflammatory cells and more epithelium in HBO; no difference in angiogenesis
Chen/2018		2.5	120	once a day	minimum 20	postburn pain score lower in HBO
Ikeda/1967		3		once or twice a day	5 to 10	6 patients died (those with 90–100% TBSA); no infection during HBO
Lamy/1970		3	60–90	once or twice a day		HBO does not alter mortality in extensive burns; fewer infections, better granulation and healing
Hart/1974	up to 24	2	90	three times on first day, then twice a day until healed	various	healing time, morbidity and mortality decreased in HBO. Healing time related to percentage TBSA
						mean healing time reduced
						mean healing time in relation to %TBSA reduced
Grossmann/1978	up to 4	2	90	every 8 h during first 24 h, twice a day thereafter		fluid requirements, healing time 2nd degree, eschar separation time, donor graft harvesting time, length of hospital stay, complications, mortality all reduced compared to non-HBO group; no paralytic ileus in severe burns and HBO, reduction in cost
Waisbren/1982						worse renal function, lower rate of non-segmented polymorphonuclear leucocytes and higher rate of bacteriemia in HBO; better healing, 75% fewer grafts in HBO
Niu/1987		2.5	90–120	2–3 during 1st 24 h; once a day thereafter		fluid loss reduced, earlier re-epithelization, overall mortality same as controls, less though in high-risk group
Cianci/1989	about 12	2	90	twice a day		duration of hospitalization and number of surgeries reduced in HBO
Cianci/1990	about 12	2	90	twice a day		duration of hospitalization, cost of burn care and number of surgeries reduced in HBO
Hammar-lund/1991	1.5; 10.5; or 21.5	2.8	60	three times a day	3	at day 6: less exsudation, less hyperemia, wound size reduced in HBO; no significant effect on complete epithelization
Brannen/1997	up to 24; mean: 11.5; one third within 8	2	90	twice a day	minimum 10; maximal 1 per % TBSA	no difference in number of surgeries, duration of hospitalization or mortality; less fluid loss (mentioned in discussion); no data provided about the subgroup with early HBO application
Niezgoda/1997	2	2.4	90	twice a day	6	exsudation, hyperemia, wound surface reduced in HBO; no effect on epithelization
Chong/2013	max 15	2.4	90	twice within 22 h	2	no effect of HBO on inflammatory markers IL Beta, 4, 6 and 10; significantly lower rate of positive bacterial cultures (staph aureus, pseudomonas).
Rasmussen/2015	0.1	2.4	90	one (mean crossover interval 37 days)	1	HBO attenuates central sensitation by thermal injury (pin-prick test, thermal threshold, mechanical threshold; seondary hyperalgesia); no peripheral anti-inflammatory effect.
Wahl/2019	0.1	2.4	90	one	1	after one single HBO long-lasting reduction of pain sensitivity surrounding injured area; immediate mitigating effect, long lasting preconditioning effect on hyperalgesia

**Table 3 medicina-57-00049-t003:** Animal experiments (*n* = 76), descriptive statistics.

Feature	Number of Experiments (Percentage)/Mean ± SD (Range)
species	
rat	44 (57.9%)
guinea pig	20 (26.3%
mouse	4 (5.3%)
dog	3 (4.0%)
pig	2 (2.6%)
rabbit	2 (2.6%)
hamster	1 (1.3%)
animals per experiment	19.1 ± 29.4 (2–250)
TBSA (percent)	21.2 ± 22.8 (0–75)
burn thickness	
partial thickness	43 (56.6%)
full thickness	34 (44.7%)
superficial	1 (1.3%)
not provided	3 (3.9%)
hours since injury	8.8 ± 29.6 (0.1–192)
ATA	2.5 ± 0.39 (2.0–3.0)
duration of HBO session	73.2 ± 20.2 (45–120)
HBO sessions per day	1.5 ± 0.9 (0.5–4)
total number of HBO sessions	17.1 ± 26.8 (0–138)

**Table 4 medicina-57-00049-t004:** Volunteers (*n* = 4) and patients (*n* = 11), descriptive statistics.

Feature	Number of Experiments (Percentage)/Mean ± SD (Range)
patients per clinical experimental group	65.8 ± 111.9 (2–402)
study design	
controlled	10 (66.7%)
randomized	5 (33.3%)
TBSA (percent)	35.2 ± 14.2 (13–65)
burn thickness	
partial thickness	15 (100.0%)
full thickness	15 (100.0%)
hours since injury	18.7 ± 7.4 (4–24)
ATA	2.2 ± 0.4 (2.0–3.0)
duration of HBO session	92.7 ± 8.8 (85–120)
HBO sessions per day	1.8 ± 0.4 (1–2)
total number of HBO sessions	9.8 ± 7.6 (2–20)

## Data Availability

Data sharing not applicable. No new data were created or analyzed in this study. Data sharing is not applicable to this article.

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
