# Peer review of "The History and Development of Hyperbaric Oxygenation (HBO) in Thermal Burn Injury"

_medicina, 2021, doi:10.3390/medicina57010049_

Round 1

Reviewer 1 Report

This is a very good organized review about an important and interesting topic.However I do have some remarks and suggestions:

the conclusion in the abstract differs a little from the conclusion in the manuscript, this has to be corrected. However the conclusion is also not  according to the findings in the review:

to draw a conclusion about patient care in burns one has especially to look to studies with patients with more or less the same outcome parameters: mortality,woundhealing time,length of stay ,scar quality and quality of live.

when we look to the patient studies from the last 30 years, there are only four of them , of these 4 only one is randomized!
all 4 studies did not show an better performance of HBO therapy for the mentioned parameters, or did even not have these parameters as outcome.

so in my opinion the conclusion of the review should be that in the last 30 years ,patient studies did not show any evidence of a positive influence of HBO therapy on the results of burncare in humans.

Author Response

Reviewer 1

Thank you for your meaningful comments.

We added a passage highlighting the necessity for relevant outcome parameters in future clinical studies on HBO in burn treatment and underlined the fact that only two randomized and two further matched-pair clinical studies have been done. The lack of explicit clinical results of HBO in burns is mentioned. The conclusions now do match those in the abstract.

Reviewer 2 Report

Manuscript ID: medicina-1063166

Title : The History and Development of Hyperbaric oxygenation 2 (HBO) in Thermal Burn Injury

The authors have presented a comprehensive synopsis of the experimental and clinical work that has been done on HBO in burns. The presentation of the results is interesting, it reads well although it is quite detailed, and the discussion adds a summary perspective on the results. The tables are also informative.

Specific comments/details

Please check the reference list, it seems like number 21-24 are placed over two references (21-22) and the rest have been shifted along.

Page 2, line 71, serendipidous, please check the spelling.

Page 17, line 401, imitation, please check the spelling.

Table 3, please consider to change the decimal separator (burn-HBO hours and ATA) from comma to period.

Author Response

Reviewer 2

Thank you for your valuable suggestions

We have checked and improved the spellings.

Round 2

Reviewer 1 Report

Thank you for the corrections and new setup off the conclusion.

now it makes more sense and is more representative for the results of the reviewed literature